# Internal Coating of Ureteral Stents with Chemical Vapor Deposition of Parylene

**Sara Felicitas Bröskamp [1], Gerhard Franz [1,\*] and Dieter Jocham [2]**

1   Department of Applied Sciences and Mechatronics, Munich University of Applied Sciences,
    D-80335 Munich, Germany; sara-broeskamp@web.de
2   Klinik für Urologie, University Hospital Schleswig-Holstein, Campus Lübeck, 160 Ratzeburger Allee,
    D-23538 Lübeck, Germany; dieter-jocham@uksh.de
\*   Correspondence: gerhard.franz@hm.edu

**Abstract:** Ureteral balloon catheters and ureteral stents are implanted in large quantities on a daily basis. They are the suspected cause for about a quarter of all the nosocomial infections, which lead to approx. 20,000 deaths in Germany alone. To fight these infections, catheters should be made antibacterial. A technique for an antibacterial coating of catheters exhibiting an aspect ratio of up to 200 consists of a thin silver layer, which is deposited out of an aqueous solution, which is followed by a second step: chemical vapor deposition (CVD) of an organic polymeric film, which moderates the release rate of silver ions. The main concern of the second step is the longitudinal evenness of the film. For tubes with one opening as balloon catheters, this issue can be solved by applying a descendent temperature gradient from the opening to the end of the catheter. An alternative procedure can be applied to commercially available ureteral stents, which exhibit small drainage openings in their middle. The same CVD as before leads to a longitudinal homogeneity of about $\pm 10\%$—at very low costs. This deposition can be modeled using viscous flow.

**Keywords:** plasma-enhanced chemical vapor deposition; xylylene; parylene; ureteral stent; viscous flow

## 1. Introduction

Poly-*p*-xylylene (PPX), with its trivial name parylene, is an organic polymer that is used as a versatile coating material to protect sensitive, even very rugged, surfaces in hostile environments [1]. Its permeability against liquids and gases is second to none where organic polymers are concerned and just one order of magnitude poorer than inorganic films made of $SiO_2$ [2,3]. As a result, very thin fractions of a μm (up to some μm) are sufficient for long-term protection. Because PPX belongs to the group of organic polymers which passed the approval of the FDA, it has been in use as a coating film on implantates for human beings for more than 40 years [4,5]. This quality especially attracted our interest, and we deposited very thin films of parylene on the inner and outer surface of urethral balloon catheters and ureteral stents to add antibacterial activity to them [6].

Ureteral balloon catheters and ureteral stents are used on a daily basis in large quantities. Almost every stationary patient who undergoes surgery has a catheter implanted as the first step. For patients with incontinence in an advanced state, it is one means to sustain individual mobility. However, they are the suspected cause for about a quarter of all nosocomial infections [7]. This collateral damage is accepted worldwide. To fight these infections, catheters have to be made antibacterial.

Catheters are implanted through the urethra. When the tip reaches the bladder, the catheter is fixed in the bladder by blocking with a balloon for balloon catheters, or with a logarithmic spiral for ureteral stents (so-called pigtails). The ureteral stent is

pushed further forward through the ureter until the tip reaches the renal pelvis, where it is mechanically fixed by a second pigtail. In both cases, the outer wall of the implant is in intimate contact with the hollow human tube over the whole length. The flow of urine is intended to happen through the artificial capillary, since the flow resistance across the capillary is very low compared with the second possibility of flow through the annular clearance between human tube and the implant. However, this flow is not impossible. To avoid preterm and instantaneous replacement of an ureteral stent, which is caused by incrustation, the ureteral stent is equipped with several drainage openings. If the flow resistance through the hollow tube is enlarged by incrustation, the urine is drained through these holes and flow happens then through the annular clearance—at a lower level (Figure 1).

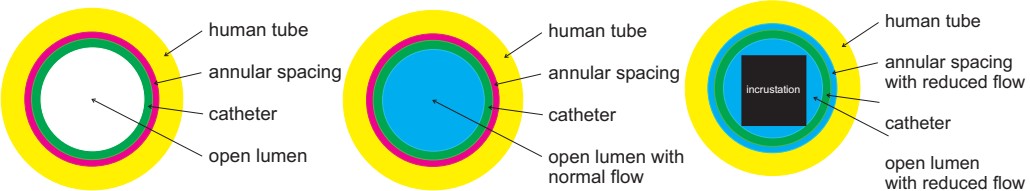

**Figure 1.** LHS—setup of the catheter in the human tube, middle—normal flow (blue), RHS—flow when blocked (blue).

Over the course of our investigations to improve uriniferous catheters with an antibacterial function, we chose silver as the bacteriostatic reagent. Coating the interior wall of a catheter with a film that is highly long-term stable (i.e., one year or longer) requires chemical vapor deposition (CVD). All the sol-gel processes or impregnations cannot yield a sufficient result (the inner diameter is just a few mm, and the viscosity of film-building gels is far too high). Because the substrate cannot be exposed to temperatures higher than 100 °C, inorganic polymers ($SiO_2$, etc.) are out of the question, and from the innumerable organic films, only a film with an FDA approval is qualified for our purposes. Since the required temperature for codeposition with a silver organic compound would exceed the temperature limit of 100 °C, we chose a two-layer sandwich of silver/silver oxide and parylene. Whereas the silver acts as antibacterial reagent, its release rate can be moderated by a very thin top layer of parylene. This coating also improves the tribological properties and, last but not least, also hampers the proceeding of bacteria [6].

The silver film is deposited in a static process out of an aqueous solution, which does not significantly deplete silver ions during the coating, and is applied to both types of catheters. The process is described elsewhere [8–12]. Briefly, the Tollens reaction is applied by which silver nanoparticles (NPs) are generated by reduction of $Ag^+$ ions by several saccharides [10,13].

For the subsequent deposition with the parylene film, however, chemical vapor deposition (CVD) is inevitable. We used the Gorham process (Figure 2).

A three-dimensional dimeric species (di-*p*-xylylene, DPX)—which contains two ethylene bridges in each *p*-position—is cracked to form the monomer (*p*-xylylene radical, MPX), which forms one-dimensional polymeric chains of *poly-p*-xylylene (PPX) [14]. This reaction can occur either in the gas (volume polymerization) or on the cold surface (surface polymerization), but only below the ceiling temperature, which marks the upper temperature limit for deposition.

Moreover, for a wafer, which is entirely exposed to a vapor of constant density, film growth is locally constant and only a linear function of time. Along the interior of a narrow tube with one opening, however, growth causes a significant loss of the film-building species and, therefore, the layer thickness decreases in an inward direction. As a longitudinally equal layer is required to ensure a constant long-term $Ag^+$ release rate, one of the possible solutions is a counteracting temperature profile. High temperature leads to a low deposition rate and vice versa [15,16].

**Figure 2.** The dimeric species (di-*p*-xylylene, DPX, and LHS)—which contains two ethylene bridges in each *p*-position—acts as precursor and is cracked to form the monomer (*p*-xylylene radical, MPX, and middle), which forms one-dimensional polymeric chains of *poly-p*-xylylene (PPX, RHS).

Whereas this procedure has been successfully applied for balloon catheters, the ureteral stents offer an alternative possibility because they are delivered with a series of consecutive drainage openings and two openings at either end of the tube with a total length of 200 mm. To investigate the influence of the number, diameter and distance of these holes on the longitudinal homogeneity of the deposited layer, we generated a series of ureteral stents with drilled holes. The layer-forming vapor can enter the tube at the two openings and the additional holes. Because the configuration of the reactor is asymmetric, where the inlet (entrance) of the layer-forming vapor and its outlet (pumping port) is concerned, the gradient of the layer-forming has to be measured in a first step. The second step consists of depositing a film with and without additional holes between the two openings at either end, and the third step is modeling the resulting layer thicknesses. If this method yields coatings which are sufficient for long-term control of silver release, this could cut the costs of the first method significantly because of the lower consumption of the precursor, which forms the coating, the higher deposition rates, resulting in a shorter deposition time, and much easier handling.

For the Gorham process [14], the deposition behavior was studied as a function of temperature by several groups. Yang and Yasuda et al. demonstrated that the deposition rate drops linearly with the increasing substrate temperature and found the ceiling temperature to be at 30 °C [17,18]. Already at our preliminary trials during our extensive work concerning this issue [16], we found that the deposition rate plateaued at very low deposition rates, which, in turn, increased the ceiling temperature and made it difficult to define this temperature precisely. We found the ceiling temperature to be $68 \pm 2$ °C [16].

Normally, very thin layers (below 50 nm in thickness) are impossible to measure. However, we solved this problem by depositing an aluminum layer on a microscope slide followed by the deposition of parylene N at temperatures between 30 °C (the previous ceiling temperature) and 80 °C, which covered the ceiling temperature of $68 \pm 2$ °C, which was evaluated by our group. Protection of the aluminum surface was demonstrated by subjecting the slide to aqueous HCl since only protected aluminum layers can withstand this exposure to acid.

During these experiments to define the ceiling temperature as exactly as possible using a temperature seesaw, we again measured and simulated the deposition at room temperature in order to evaluate the competion between normal propagation by diffusion and monomeric deposition. According to Fortin [19,20], the monomers can only deposit at the end of a growing polymeric chain, thereby excluding growth in two or three dimensions.

From the mechanistic point of view, it is interesting to see whether the different rarefaction ratios ($\frac{\lambda}{2r}$) with $\lambda$, the mean free path, and $2r$, the diameter of the pipe (either the diameter of the reactor or the diameter of the tube, respectively) changes the deposition rate.

In contrast to other deposition techniques, which are carried out in vacuum, e.g., evaporation or sputtering, single molecules or reactive fragments deposit on the surface and form chemical bonds with their neighbors. In fact, the chemical nature of the substrate

influences the deposition rate. It was first pointed out by Vaeth and Jensen that PPX growth on layers of transition metals, in particular, copper and silver, starts with a certain delay [21]. From the two possible mechanisms for this obstacle, namely the lower sticking coefficient or deactivation of the landed monomers by the substrate layer, they proposed the second alternative.

From these considerations, it is obvious that this topic can be divided into two main issues, namely the following:

- Deposition of PPX on inorganic surfaces (metals, glass);
- Deposition of PPX on the internal surface of ureteral stents (here: polyurethane).

In this article, we focus on the second issue.

## 2. Experimental

### 2.1. CVD Reactor

The Gorham CVD process occurs in a reactor which is extensively described elsewhere [6,22]. Briefly, the reactor (Plasma Parylene Systems, Rosenheim, Germany) had a volume of 91 l and was flanged to an evaporator and cracker via a gate on the left, referred to as the front door (Figure 3).

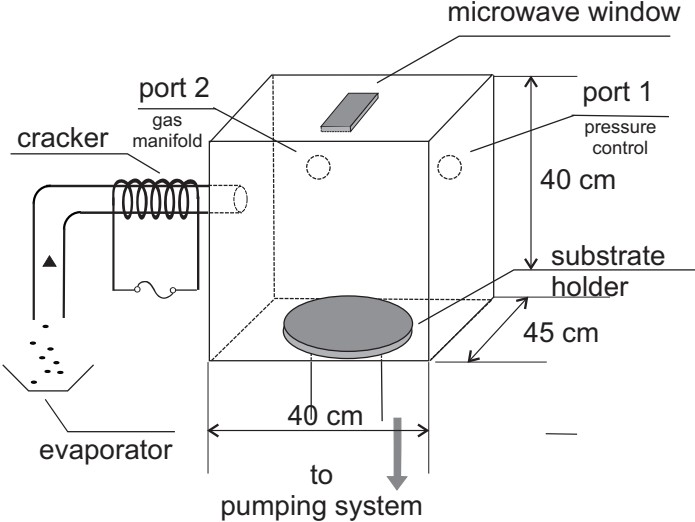

**Figure 3.** Sketch of the CVD reactor with microwave window and facilities for gas doping and pressure control. Its configuration is asymmetric with respect to gain and loss of the layer-forming vapor.

The reactor was evacuated with a rotary vane pump (Leybold 65 BCS, Cologne, Germany) through a baffle, which was maintained at −90 °C to avoid contamination of unreacted monomers. Following Gorham, a precursor was evaporated between 105 and 130 °C and thermally cracked in a cracking unit at temperatures driven at 700 °C [14]. For medical purposes, we used the solid precursor diparylene N (Plasma Parylene Systems, Rosenheim, Germany). The pressure was controlled with a heated pressure transducer (MKS Baratron 631C-01-TBFH), which ensures long-term stability. This was confirmed with our "gauge" pressure transducer B270 from MKS, which was calibrated by MKS (MKS Germany, Munich, Germany). Because of the large thermal inertia of the evaporator and the cracker, the time lag before these temperatures were reached was approximately 45 min. To avoid uncontrolled evaporation before the working temperature was reached, argon was admitted at moderate flows (80 sccm) and then suddenly cut off (drop to 2.5 sccm during the coating process), which caused a steep increase within seconds and stable evaporation [23].

### 2.2. Ureteral Stent

In contrast to the balloon catheter with an inner diameter of 1.6 mm, the ureteral stent exhibits just 1.0 mm in diameter at a length of typical 20 cm. Commercially available stents exhibit a series of small drainage holes below 0.9 mm and the two pigtails at either end. From its middle, a total of 4 holes were drilled, each of them with a diameter of 0.8 mm. The ends of this tube were either sealed or open. For PPX deposition of the ureteral stents, a carousel was placed atop the revolving plate with a capacity of 24 stents (Figure 4).

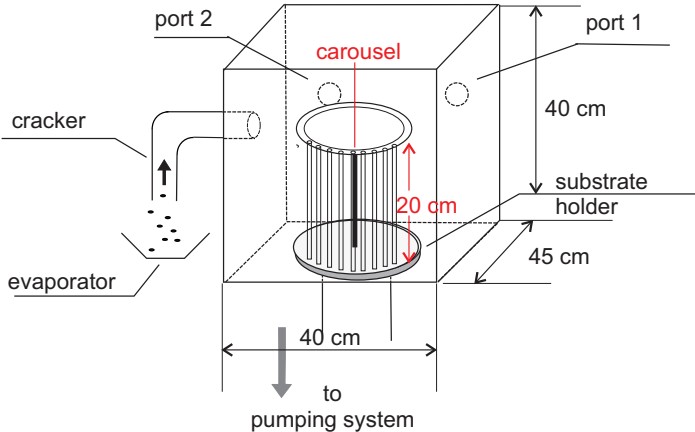

**Figure 4.** Sketch of the upgraded CVD reactor with carousel for coating of ureteral stents (maximum occupation: 20 stents). The length of the capillaries is 20 cm; they are fixed on a carousel and are vertically aligned.

### 2.3. Spatial Deposition Rate

For controlling the deposition rate across the reactor. which was asymmetric with respect to the vapor inlet and gas outlet, a three-stage device was constructed, which hosts microscope slides: four slides on the three stages for radial control, and four slides on one axial carrier (see Figure 5).

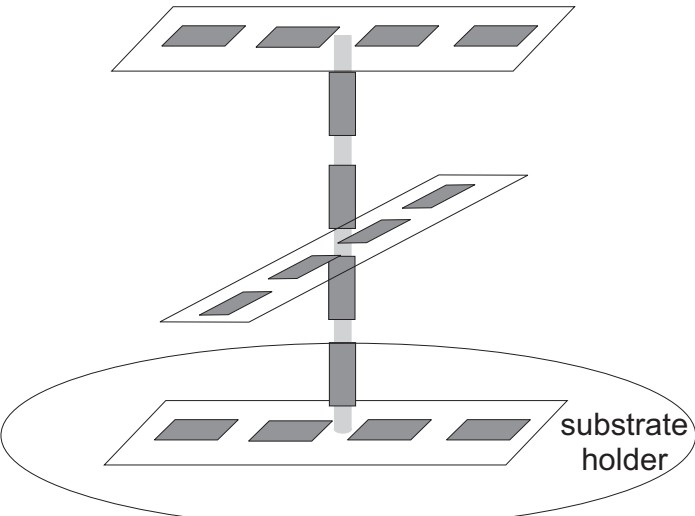

**Figure 5.** Device for controlling the spatial deposition rate across the reactor. Each of the four carrier plates hosts four microscope slides. At each trial, only one arm is subjected to coating.

Each arm was subjected to coating separately to avoid mutual influencing—coating at the upper level caused a reduction in the density of MPX, which would lead to a reduced deposition rate at a lower arm. Thickness measurements were carried out at three positions on each slide.

### 2.4. Film Thickness

Conventional mechanical profilometry with a diamond needle was carried out for these glass substrates. However, it is impossible in conjunction with soft substrates, such as the organic polymers (polyurethanes) of the urinary stents. Instead, optical measurements had to be applied. Because the refractive indices of the two materials employed in this work (the capillary wall and the polymeric layer) were almost the same, determining the thickness by intensity loss, according to Beer's law, was not possible. To measure the thicknesses of the films inside a small capillary in the present work, the optical spot had to be small compared with the radius of the curvature to avoid erroneous results. Instead, a method termed thin film interferometry, also known as reflectometry, appeared to be best suited to this scenario.

Thus, the broad-band reflectance of the capillary was recorded as a means of tracking the film thickness.

For this purpose, we employed an F20 e spectrometer (Filmetrics, Unterhaching, Germany), using a light spot with a diameter $d_s$ of 30 μm. In the case of capillary radius $r$ of 0.75 mm, the substrate could be regarded as plain ($r \gg d_s$). The reflected light was diffracted by a grating and detected using a photodiode array. By relating the recorded spectrum of the substrate with the layer in place to a previously recorded spectrum of the substrate alone, a background-corrected signal proportional to the thickness and based on the refractive index of the probed layer can be obtained. To measure the thickness on the inner side of the ureteral stents, the specimens need to be cut with a surgeons's knife.

### 3. Results

#### 3.1. CVD in the Reactor

The process pressure was fixed to 5–6 Pa (40 mTorr), resulting in a mean free path $\lambda$ of approx. 1 mm, which meant that the rarefaction ratio $\delta = 2r/\lambda$ was approximately 800 in the reactor, but equaled unity in the catheters, which led to a Knudsen number of 1. In both systems, no pressure gradients existed and therefore no convective flow. Therefore, the only mechanism to equalize differences in partial pressure (here of MPX) was diffusion.

Deposition occurred on all surfaces with a temperature below the ceiling temperature. The largest surface was the interior reactor wall. According to Figures 3 and 4, the surface of the reactor was 10,400 cm². The objects, which were subjected to the coating process, compete with the reactor surfaces and should not change the overall flow to avoid an additional depletion of the film-building reagent—well known as the loading effect. Therefore, the area ratio between the reactor surface and the objects should be small.

To measure the vertical and radial loss, a total of 16 microscope slides were inserted in the reactor with an area of 80 cm² (each slide: 20 cm²) in four consecutive runs. Compared with the inner area of the reactor of approx. $10 \times 10^3$ cm², this was a ratio of less than 1%. In the case of the ureteral stents, their area could be completely neglected (surface of one catheter is 1.5 cm²) (cf. Figures 4 and 6).

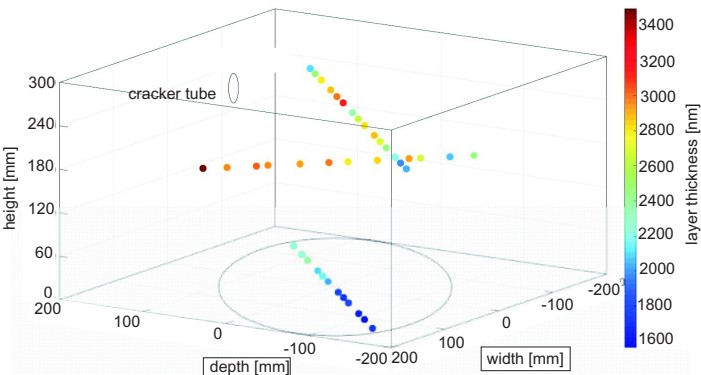

**Figure 6.** Layer thickness across the reactor. Each dot represents the averaged measurements across a microscope slide (length: 76 mm).

Numerous measurements of the resulting layer thickness along the axial (50 cm) and radial (±23 cm) dimensions (where axial is effective from the bottom at a position of 4.8 cm to the top at 31.6 cm, and radial is from −23 cm to +23 cm) resulted in the following pattern:

- The vertical distribution shows higher values at the top.
- The radial distribution shows an influence of the entrance gate at the left (at negative radii), Figure 7, which are numerically fitted with an erfc function.

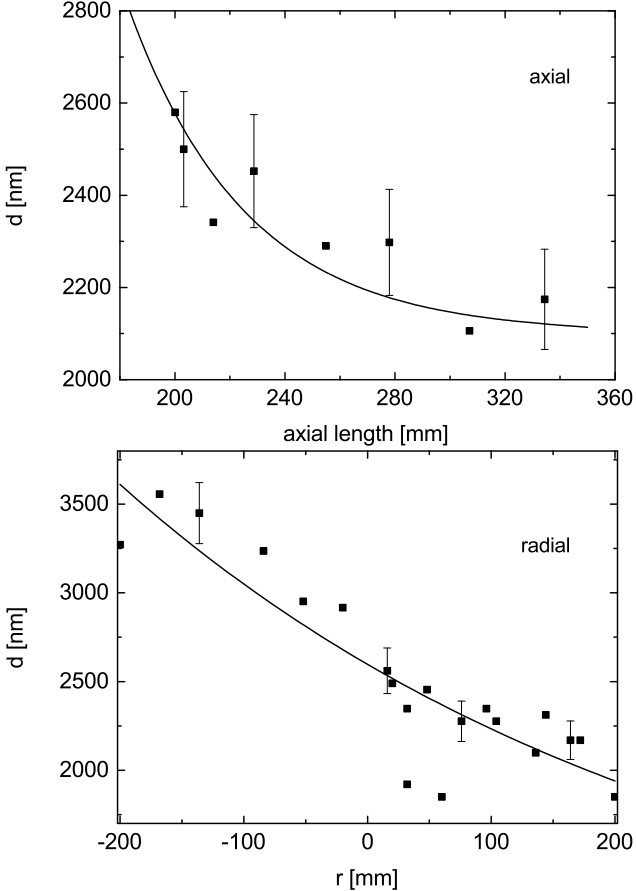

**Figure 7.** Axial and radial distribution of layer thickness in the CVD reactor, the radial measurement in the middle of the axis. Area load: < 1% of the inner reactor area. Clearly visible is the influence of the gas feed at the left (negative values of the radius). At the top, the deposition rate is higher than at the bottom. Fits with an erfc function.

From the trials with the carousel, we found an axial drop in thickness $d$ of less than 10% across a length $l$ of 30 cm of the ureteral stents ($\frac{\Delta d}{d_{max}} = 7.2\%$). A radial dependence across the inner 20 cm did not exist.

The maximum difference was, therefore, from top left to right bottom. These points were subjected to a numerical analysis with the diffusion equation with a small loss term, as shown in Equation (1):

$$\frac{\partial c}{\partial t} = D \frac{\partial^2 c}{\partial x^2} - L c, \tag{1}$$

Here, $L$ is the rate constant of the loss reaction. Equation (1) can be solved by standard methods and is the sum of the two terms in Equation (2):

$$\begin{aligned} \frac{c(x,t)}{c_0} &= \frac{1}{2} \exp\left(-x\sqrt{\frac{L}{D}}\right) \mathrm{erfc}\left(\frac{x}{2\sqrt{Dt}} - \sqrt{Lt}\right) \\ &+ \frac{1}{2} \exp\left(x\sqrt{\frac{L}{D}}\right) \mathrm{erfc}\left(\frac{x}{2\sqrt{Dt}} + \sqrt{Lt}\right) \end{aligned} \tag{2}$$

with erfc($x$), the complementary error function. The borderline cases are

- $L \gg \frac{x^2}{D}$ (losses dominate diffusion);
- $L \to 0$ (no losses).

Since the limits of the complementary error function are $\lim_{x \to \infty} \text{erfc}(x) = 0 \wedge \lim_{x \to -\infty} \text{erfc}(x) = 2$, in the case of dominating loss, the second summand vanishes in Equation (2) while, in the first summand, the exponential term dominates the erfc term to arrive at the following time-independent exponential function Equation (3):

$$\frac{c(x)}{c_0} = \exp\left(-x\sqrt{\frac{L}{D}}\right). \tag{3}$$

For $L \to 0$ (no losses), only the following diffusion term remains in Equation (2) to arrive at Equation (4)

$$\frac{c(x,t)}{c_0} = \text{erfc}\frac{x}{\sqrt{4\,D\,t}}. \tag{4}$$

The consideration above holds true for viscous flow, meaning that the mean free path $\lambda \ll \Lambda$, the diffusion length. In the range of some tens of mTorr, $\lambda$ for nitrogen is approximately 1 mm (at 20 mTorr: 2 mm with a $\sigma(N_2)$ of 41.6 Å$^2$). For an inner reactor diameter in the vicinity of 10 cm, even for a tube diameter of 10 mm, $r \gg \lambda$. $D$, as determined based on the kinetic theory of gases, is $D = {}^1\!/{}_3 \lambda \sqrt{\langle v^2 \rangle}$, where $n$ is the number density and $\langle v^2 \rangle$ is the mean squared velocity of 3750 cm$^2$/s. The exact values for the monomeric species MPX, $C_8H_8$, are unknown and so we used values that are available for benzene [24]: $\sigma = 107.5$ Å$^2$ and thermal speed $\langle v(\text{MPX}) \rangle = 0.51 \langle v(N_2) \rangle$. More precisely, $\Lambda$ has to be calculated using the random walk equation. At 20 mTorr, a reactor volume of 72 L and a flow of 10 sccm, the residence time in the reactor is 12.4 s, and $\Lambda$ is more than 3 m (304 cm).

For the maximum distance (top left to right bottom), the experiments were simulated and are shown in Figure 8, employing various diffusion coefficients and loss rates.

The diffusion coefficient is evidently of the order of $10^3$ cm$^2$/s, while the loss rate is below $0.5\,\text{s}^{-1}$, yielding a ratio of approx. $2 \times 10^3$ cm$^2$. Thus, the loss rate appears to affect the deposition rate only moderately. This small loss term well agrees with the sticking ratio values of approximately $10^{-4}$ as reported by Beach [25].

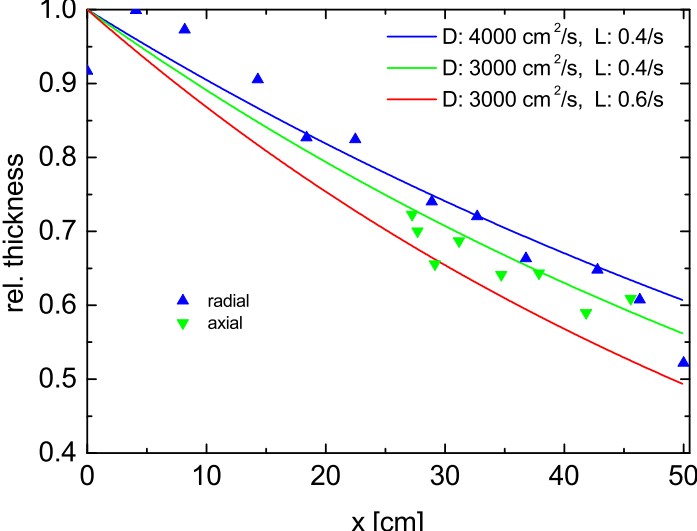

**Figure 8.** Fitting of the experimental reactor deposition data from Figure 7 with the diffusion loss Equation (4) at 20 mTorr and a 300 K surface temperature. Distance values, $x$, are relative to the cracking unit outlet.

### 3.2. CVD in the Tube

In the ureteral stents with an aspect ratio of $\frac{l}{2r} \approx 200$, the situation seemed to be different. As in the reactor itself, there was no pressure gradient along the tube; therefore, we had to take into account only diffusion.

We used three configurations to evaluate the competition between diffusion and deposition (Figure 9):

1. Both ends open;
2. Both ends closed but 4 equidistant drainage openings in the middle;
3. Both ends open and 4 drainage openings.

Since the lower entrance of the capillary had a lower vapor density (cf. Figures 4 and 6), this yielded an asymmetric behavior of the thickness, which had to be taken into account by the lower source density of 0.928 (instead of 1) at the upper entrance.

Following Equation (2) with a ratio of $\frac{D}{L} = 80 \, \text{cm}^2$, the upper graphs in Figure 9 were obtained. With this ratio, the third catheter type (open at its ends and equipped with drainage openings) was perfectly fit.

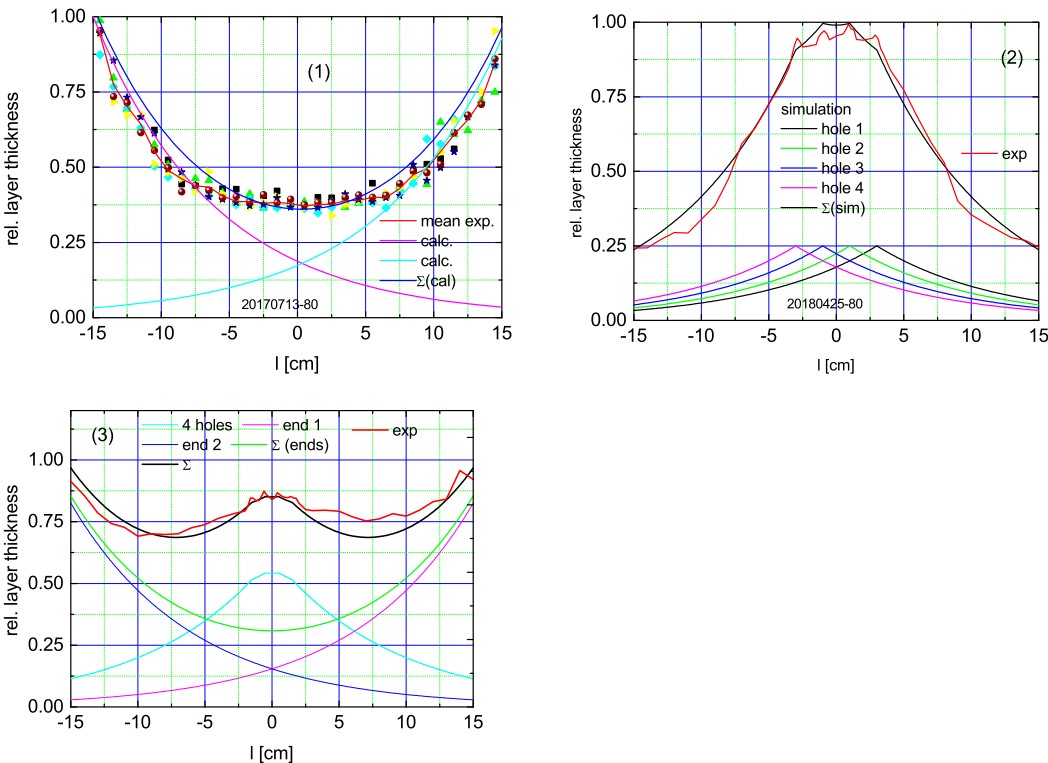

**Figure 9.** Relative thickness of the PPX layer as function of length. (**1**) Both ends open, (**2**) both ends closed, 4 drainage openings separated by 2 cm in the middle, (**3**) both ends open and 4 drainage openings. The lower vapor density at the lower entrance ($-7.2\%$ is taken into account for simulations 1 and 3. In 3, the central holes are only 1 cm apart.

The ratio $\frac{D}{L}$ in the capillaries was smaller by almost two orders of magnitude compared with the ratio in the free reactor. This was caused by the higher mean temperature of the vapor in the whole reactor—its topmost level at the cracker was 700 °C, and, according to the kinetic gas theory, $D$ scales linearly with $T \left( D = \frac{1}{3}\lambda\sqrt{\langle v^2 \rangle} \right)$ with $\lambda$, the mean free path, and $\sqrt{\langle v^2 \rangle}$, the RMS of the squared velocity. The second argument is more fundamental. In the capillary with a rarefaction ratio $\delta = 2r/\lambda$ of just unity, the application of the viscous theory is at least questionable. This was thoroughly investigated by Redka et al. [26].

Although the aspect ratio $\frac{l}{2r}$ is extremely high, which should cause a steep exponential drop in the layer thickness, utilizing the drainage openings as an additional vapor source

mitigated this issue significantly. As a result, the deviations around the mean value were reduced to $\pm 10\%$—a breakthrough in cheap, antibacterial urinary stents.

## 4. Conclusions

Depositing a film with equal thickness on the interior side of a narrow capillary remains a challenge. For tubes with one opening, coatings can be fabricated, using a thermal seesaw, which forces back the deposition in the entrance region of the tube, thereby counteracting the drop in vapor density in an inward direction. This leads to a film thickness that is longitudinally homogeneous. This technique ends in a high-end catheter; however, it is costly to realize. An application is balloon catheters for urinal purposes.

Urinary stents with their aspect ratio ($\geq 200$) are even less suited for even coating. However, we can make use of the very tiny holes by which the urinary stent is perforated at equal distances in the middle range, so-called drainage openings, which are perfectly suited to counteract the density drop of the layer-generating vapor in an inward direction by acting as a second vapor source—located just in the region of lower vapor density inside the tube.

To summarize, this is huge progress. Although the method with the temperature seesaw is very innovative, it suffers from the fact that only four catheters can be deposited in one run. For a device such as the balloon catheter that should cost only a few cents, this is a prohibitive objection.

Therefore, the improvement of the coating of urinary stents is of significant importance. Moreover, a balloon catheter can be replaced by a skilled nurse; for replacement of a urinary stent, however, a medical doctor is always required and the surgery is performed under toponarcosis. This exceeds the costs of the first action by at least two orders of magnitude.

**Author Contributions:** S.F.B. designed and performed the experiments, S.F.B. and G.F. analyzed the data and wrote the paper; D.J. supervised the whole work. All authors have read and agreed to the published version of the manuscript.

**Funding:** Financial support of the Federal Secretary of Economy under project number KF257-5103 CR4 is greatly appreciated.

**Institutional Review Board Statement:** Not applicable.

**Informed Consent Statement:** Not applicable.

**Data Availability Statement:** All experimental data are stored at Department 06 of Munich University of Applied Sciences.

**Conflicts of Interest:** The authors declare no conflict of interest.

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
