# Peer review of "Internal Coating of Ureteral Stents with Chemical Vapor Deposition of Parylene"

_coatings, doi:10.3390/coatings11060739_

Round 1
Reviewer 1 Report
Introduction must be improved for better highlighting the originality of the present approach.
Results about the antibacterial efficiency of the coating must be added.
Author Response
We have altered the direction of impact of this paper. Its previous approach was to report on the improvements in coating the inner wall of ballon catheters AND urinary stents.
Although there is a progress with our seesaw, we dropped this passage completely, just concentrating on the effective coating of tubes which are used as urinary stents. For this end, we simply renewed the Introduction and added significant parts.
However, this progress in clarity should not be sacrificed again by reporting on
qualities which belong to a completely other field, here: the biological or antimicrobial efficiency of the catheters. Therefore, we consider a very soon reporting on these breath-taking properties in a journal which is devoted to medical/microbiological issues. For example, in the past, we have published three papers in Biointerphases [1] - [3].
[1] H. Heidari Zare, St. Sudhop, F. Schamberger, and G. Franz, Microbiological investigation of an antibacterial sandwich layer system, Biointerphases 9, 031002 (2014).
[2] H. Heidari Zare, O. D¨uttmann, A. Vass, G. Franz, and D. Jocham, Silver ions
eluted from partially protected silver nanoparticles, Biointerphases 11, 031002 1. 9 (2016).
[3] H. Heidari Zare, V. Juhart, A. Vass, G. Franz, and D. Jocham, Efficacy of
silver/hydrophilic poly(p-xylylene) on preventing bacterial growth and biofilm
formation in urinary catheters, Biointerphases 12, 011001 1 . 10 (2017).
Reviewer 2 Report
On request of Coatings, I have revised the manuscript titled “Internal coating of ureteral stents with chemical vapor deposition of parylene”, by Sara Felicitas Bröskamp, Gerhard Franz and Dieter Jocham.
After reading only the abstract, it was not clear to me when the authors were talking about strategies already used and therefore about the background and when instead they were talking about what was developed in the present study. I thus recommend more clarity on what the authors actually did, starting with the abstract.
Reading the Introduction and the rest of the manuscript, I realized that the coating system developed here is the same that Gerhard Franz, Florian Schamberger, Hamideh Heidari Zare, Sara Felicitas Bröskamp and Dieter Jocham, developed and studied in their work in 2017 ( Ref. 6). The present work, especially for the experimental part which is however very poor and sparse, seems to be just a summary of the previously published work (Ref. 6).
The materials developed, that is the coating consisting of double silver layers protected with parylene as an organic polymer capable of releasing silver ions, are the same already described in the 2017 work and have also been prepared by applying the same techniques used then. Furthermore, the strategies adopted to overcome the main problems concerning the preparation and deposition of this coating are identical to those already presented. Some Figures, such as Figures 2 and 3 are even the same Figures 4 and 3 that are present in their previous work (Ref. 6).A real experimental part with the description of the laboratory procedures is missing.
Furthermore, the English language needs careful revision since there are many grammatical errors, often the tenses of the verbs are not adequate and the sentences often lack a subject or object complement.
Author Response
Yes, it is this method of depositing a sandwich layer of silver and parylene which is described in precedent papers. Therefore, we have used some figures of these papers to describe the reactor and some properties of our films.
We succeeded in depositing films with equal thicknesses on the interior side of
capillaries which are open just on one end (balloon catheters), the last of these papers which has been cited by the reviewer was by S. Br¨oskamp, D. Redka, A. M¨ohlmann, G. Franz, and D. Jocham [1].
In both cases (balloon catheters and ureteral stents), the propagation of the filmbuilding monomer inside the tube happens by diffusion only. No convection because there is no field. In the first case of the balloon catheters with an aspect ratio below 100, viscous flow is assumed. But for the urinary stents, we are touching the Knudsen regime. Hence we had to check whether we could work with the first assumption of viscous flow—which is much easier.
Following your advice, we concentrated on preparative aspects for urinary stents. Their aspect ratio (≥ 200) is hardly suited for even coating. However, we could could make use of the very tiny holes by which the urinary stent is perforated at equal distances—so-called drainage openings. They act as additional vapor source
and equalize the steep drop of film-building monomers in inward direction to the middle of the catheter. As result, we could model the deposition with viscous flow (without mentioning it).
For our project for antibacterial catheters, this is a huge progress. Although the
method with the temperature seesaw is very innovative it suffers from the fact that only four catheters can be simultaneously deposited in one run. For a device that may not cost more than $1, this is a prohibitive objection.
Reviewer II demanded to focus on the actual actions in this work, beginning with the abstract. Well, we did so, over the whole text, and also in the Conclusion.
Because we focus now on the coating process for the ureteral stents solely, we must compare the different behavior between balloon catheters and ureteral stents in the Introduction (p. 4, line 89). And we dropped the actual version of the temperature seesaw for balloon catheters. Thanks to his proposals, we consider the paper now more focused with the clear statement that another challenge for fabricating antibacterial ureteral stents has been mastered.
The English has been checked by a native speaker. Especially the change of tenses through the text has been recognized and corrected.
[1] S. Br¨oskamp, D. Redka, A. M¨ohlmann, G. Franz, and D. Jocham, Chemical
vapor deposition of poly-p-xylylene in narrow tubes, AIP Advances 7, 075005
(2017).
Reviewer 3 Report
Thank you for submitting the good manuscript.
I have few questions and comments.
Has any research been done on the durability or realiability of the currently applied coating technique? I think that It would be better to include the results of the durability or realiability of the currently coated stents.
I wonder the changes in terms of mechanical strength by the proposed method.
In the case of ureter stents, it would be better to mention whether uniform performance can be achieved for curved shapes, as they include both sides of the pig tail shape and curved tubes.
Author Response
The durability of the coated capillaries has been investigated in comparable investigations during the last years (see our publications in Biointerphases [1] − [3]). The antimicrobial efficiency has been proven for a maximum of 28 days. The next publication is scheduled in this or comparable journals. Since other students are involved in this issue, we avoided to mingle these two groups by publishing one paper with a double focus: preparation of these tubes (more engineering) and their antibacterial efficiency (more medical/biological)—as in the first version of this paper: progress in balloon catheters AND ureteral stents.
No, we have not measured the thickness of our coating film in the pigtail itself.
When you imagine the difficulties of measuring the layer thickness on a strong curvature (inner diameter of less than 1 mm, as we had to do, each stent has been measured thirty times, and my students have measured over weeks!), these are even more pronounced in the pigtail.
[1] H. Heidari Zare, St. Sudhop, F. Schamberger, and G. Franz, Microbiological investigation of an antibacterial sandwich layer system, Biointerphases 9, 031002 (2014).
[2] H. Heidari Zare, O. D¨uttmann, A. Vass, G. Franz, and D. Jocham, Silver ions
eluted from partially protected silver nanoparticles, Biointerphases 11, 031002 1. 9 (2016).
[3] H. Heidari Zare, V. Juhart, A. Vass, G. Franz, and D. Jocham, Efficacy of
silver/hydrophilic poly(p-xylylene) on preventing bacterial growth and biofilm
formation in urinary catheters, Biointerphases 12, 011001 1 . 10 (2017).
Round 2
Reviewer 1 Report
The authors have addressed or explained enough the initial concerns regarding the manuscript.
Author Response
Thank you for your comment.
Reviewer 2 Report
The authors have modified their paper, thus improving the quality of the manuscript and above all the language, as well as have focused on certain aspects as I suggested. Now the abstract is clear and well written.
I have two residual requests that authors should address to make their manuscript suitable for publication on Coatings.
- The original Figure 2, Eq. 1 now it has become only Equation (1). I disagree, because it is a real chemical process, so it must be presented as a Scheme 1 and needs a caption.
- Authors should modify Scheme 1 and Figure 3 (which will become Figure 2) to differentiate this manuscript from their previously published work.
Once these two points have been solved, the work can be published.
Author Response
Dear Sir or Madam,
thank you for your comments and requests! I feel and know that the manuscript has won in clarity AND impact.
To your proposals:
I have reverted Eq. (1) back to Fig. 1; the numbers of all equations are reduced by 1, the numbers of the following figures have been enlarged by 1.
I have redrawn now Fig. 1. I have added a furnace symbolizing the cracker unit on the LHS, have sharpened port 1 and port 2 by adding "gas manifold" and "pressure control".
In line 236, I have used larger brackets.
Thank you again,
Gerhard Franz